

# Unanticipated discovery of two rare gastropod molluscs from recently located hydrothermally influenced areas in the Okinawa Trough

Chong Chen[1], Hiromi Kayama Watanabe[2,3,4], Junichi Miyazaki[1,3,4] and Shinsuke Kawagucci[1,3,4]

[1] Department of Subsurface Geobiological Analysis and Research, Japan Agency for Marine-Earth Science and Technology, Yokosuka, Kanagawa, Japan
[2] Department of Marine Biodiversity Research, Japan Agency for Marine-Earth Science and Technology, Yokosuka, Kanagawa, Japan
[3] Research and Development Center for Submarine Resources, Japan Agency for Marine-Earth Science and Technology, Yokosuka, Kanagawa, Japan
[4] Project Team for Development of New-generation Research Protocol for Submarine Resources, Japan Agency for Marine-Earth Science and Technology, Yokosuka, Kanagawa, Japan

## ABSTRACT

**Background**. The deep-sea hydrothermal vent is one of the most 'extreme' environments in the marine realm. Few species are capable of inhabiting such ecosystems, despite extremely high productivity there supported by microbial chemosynthesis, leading to high biomass and low species richness. Although gastropod molluscs are one of the main constituents of megafaunal communities at vent ecosystems, most species belong to several typical families (e.g., Provannidae, Peltospiridae, Lepetodrilidae) specialised and adapted to life at vents.

**Methods**. During recent surveys of Okinawa Trough hydrothermal vent systems, two snails atypical of vent ecosystems were unexpectedly found in newly discovered hydrothermally influenced areas. Shell and radular characteristics were used to identify the gastropods morphologically.

**Results**. One species was a vetigastropod, the calliostomatid *Tristichotrochus ikukoae* (*Sakurai, 1994*); and the other was a caenogastropod, the muricid *Abyssotrophon soyoae* (*Okutani, 1959*). Both gastropods were previously only known from regular non-chemosynthetic deep-sea and very rare—only two definitive published records exist for *T. ikukoae* and three for *A. soyoae*. The radula formula of *Tristichotrochus ikukoae* is accurately reported for the first time and based on that it is returned to genus *Otukaia*. For both species, barcode sequences of the cytochrome *c* oxidase I (COI) gene were obtained and deposited for future references.

**Discussion**. These new records represent the second record of calliostomatids from vents (third from chemosynthetic ecosystems) and the third record of muricids from vents (tenth from chemosynthetic ecosystems), and extend the distribution of both species to the southwest. Neither family has been recorded at chemosynthetic ecosystems in the western Pacific. Both were from weakly diffuse flow areas not subject to high temperature venting but were nevertheless associated with typical vent-reliant taxa such as *Lamellibrachia* tubeworms and *Bathymodiolus* mussels. These new records show that these species are capable of tolerating environmental stress associated with

Corresponding author
Chong Chen, cchen@jamstec.go.jp

weak hydrothermally influenced areas, despite not being vent endemic species, adding to the list of known vent/non-vent species intersections. This signifies that such weakly influenced areas may provide key habitats for them, and that such areas may play a role in the evolution of biological adaptations to 'extreme' chemosynthetic ecosystems.

# INTRODUCTION

Deep-sea hydrothermal vents exhibit increased water temperature, higher heavy metal concentrations, and raised concentrations of substances acting as energy sources for chemosynthesis (such as $H_2$, $CH_4$, $H_2S$) compared to the surrounding water (*Van Dover, 2000*; *Petersen et al., 2011*). Due to such 'extreme' environmental conditions, few species are capable of inhabiting such ecosystems, despite extremely high productivity there for the deep-sea supported by microbial chemosynthesis (*Grassle, 1987*). As a result, although vent ecosystems support biomass comparable even to that of shallow water coral reefs, the macro- and megafaunal communities there are characterised by a low species richness (*Lutz & Kennish, 1993*; *Van Dover & Trask, 2000*).

Although gastropod molluscs are one of the main constituents of megafaunal communities at hydrothermal vent ecosystems, species present mostly to belong to one of several typical families (e.g., Provannidae, Peltospiridae, Lepetodrilidae) specialised and adapted to life at vent and other chemosynthetic ecosystems such as hydrocarbon seeps and organic falls (*Sasaki et al., 2010*). Here, we report new records of two gastropod species, *Tristichotrochus ikukoae* (*Sakurai, 1994*) (Vetigastropoda: Calliostomatidae) and *Abyssotrophon soyoae* (*Okutani, 1959*) (Caenogastropoda: Muricidae), from two new diffuse flow hydrothermal sites discovered using the multibeam echo-sounding (MBES) method (as outlined in *Nakamura et al. (2015)*). Both species are exceedingly rare endemics of Japan, with only two published records each.

Calliostomatidae is an exclusively marine family of usually grazing gastropods which comprises about 300 extant species (*Marshall, 1995*; *Vilvens, 2014*; *Marshall, 2016*). Only two species have been recorded from chemosynthetic ecosystems, one from vent and one from seep (*Sasaki et al., 2010*). Muricidae is a large family of carnivorous and generally predatory marine caenogastropods comprising about 1,600 living species (*Merle, Garrigues & Pointier, 2011*). A total of seven muricids have been recorded from hydrocarbon seeps, but none from vents (*Sasaki et al., 2010*) and two species (*Enixotrophon carduelis* (Watson, 1882) and *E. obtusus* (*Marshall & Houart, 2011*) were recorded from Rumble V Volcano which is a hydrothermally influenced area on the Kermadec Ridge (*Marshall & Houart, 2011*). The present study, therefore, presents the second record of Calliostomatidae and the third record of Muricidae from hydrothermal vent ecosystems.

The two species newly recorded from the vents herein are unlikely to be vent endemics but are clearly interacting with vent environment and other vent endemic fauna. These

records add to the intersection of vent endemic and non-vent endemic species in the peripheral diffuse flow environment, where non-endemic species have often been recorded due to the much less stressful environment compared to focused flow venting (reviewed in *Levin et al., 2016*).

## MATERIALS & METHODS

Potential signals of hydrothermal activity were detected in the Okinawa Trough by detecting acoustic water column anomalies, probably derived from $CO_2$ bubbles discharging from vents, using an EM122 (Kongsberg Maritime, Kongsberg, NO) MBES system on R/V *Yokosuka* (cruises YK14-16 and YK16-07), as outlined in *Nakamura et al. (2015)*. The signals were then ground-truthed with Remotely Operated Vehicle (ROV) *KAIKO* (with vehicle *Mk-IV*) dives on-board R/V *Kairei* during cruises KR15-16 (*Kawagucci et al., 2015*; *Makabe et al., 2016*) and KR16-16 (*Miyazaki et al., 2016*; *Makabe et al., 2017*). A suction sampler equipped on the ROV *KAIKO* was used to collect the gastropods. A digital camera mounted on the same ROV was used to shoot the *in situ* observation images.

Upon recovery on to the research vessel, the specimens were fixed and stored in 99% ethanol for subsequent studies. Amplification and sequencing of the cytochrome *c* oxidase I (COI) barcode gene was done using a piece of foot using the universal primer pair LCO1490 and HCO2198 (*Folmer et al., 1994*). Molecular methods follow that outlined in *Chen, Watanabe & Ohara (2016)*. After taking tissue snips, soft parts were extracted from the specimens after rehydrating in MilliQ. The radula was dissected out from each specimen under a dissecting microscope (Olympus SZ1; Olympus, Shinjuku, Tokyo, Japan) and cleaned by washing in diluted commercial bleach followed by two washes, in MilliQ water and 99% ethanol each. The cleaned radulae were mounted on SEM stubs with carbon tapes and examined uncoated at 15kV with a table top scanning electron microscope (SEM; Hitachi TM-3000; Hitachi, Tokyo, Japan).

## RESULTS AND DISCUSSION

### Detection and discovery of hydrothermal sites

One acoustic anomaly (28°26.1′N, 128°11.5′E; Fig. 1) was detected in the northern Okinawa Trough near Tokara Islands, in the 'Higashi-Ensei' area (*Makabe et al., 2017*) east-northeast of Minami-Ensei Knoll (*Hashimoto et al., 1995*). This site was explored on dive #724 of ROV *Kaiko*, which discovered a weakly active hydrothermal vent site mostly with diffuse flow areas. This site was 955–1,180 m deep, dominated by dense *Lamellibrachia* tubeworm bushes grown over by the barnacles *Ashinkailepas seepiophila* Yamaguchi, Newman and Hashimoto, 2004 and *Leucolepas longa* Southward and Jones, 2003, and was named 'Fukai' (Fig. 2A; for details see (*Miyazaki et al., 2016*; *Makabe et al., 2017*)). Other fauna typical of Okinawa Trough hydrothermal vents (*Watanabe & Kojima, 2015*), such as the squat lobster *Shinkaia crosnieri* Baba & Williams, 1998, the vesicomyid clam *Akebiconcha kawamurai* Kuroda, 1943, and the mussel *Bathymodiolus platifrons* Hashimoto & Okutani, 1994 were also present in this site. Another signal (25°4.5′N, 124°31.0′E; Fig. 1) was located about 1,950–1,990 m deep on the northern slope of Tarama Hill in the southern Okinawa Trough,

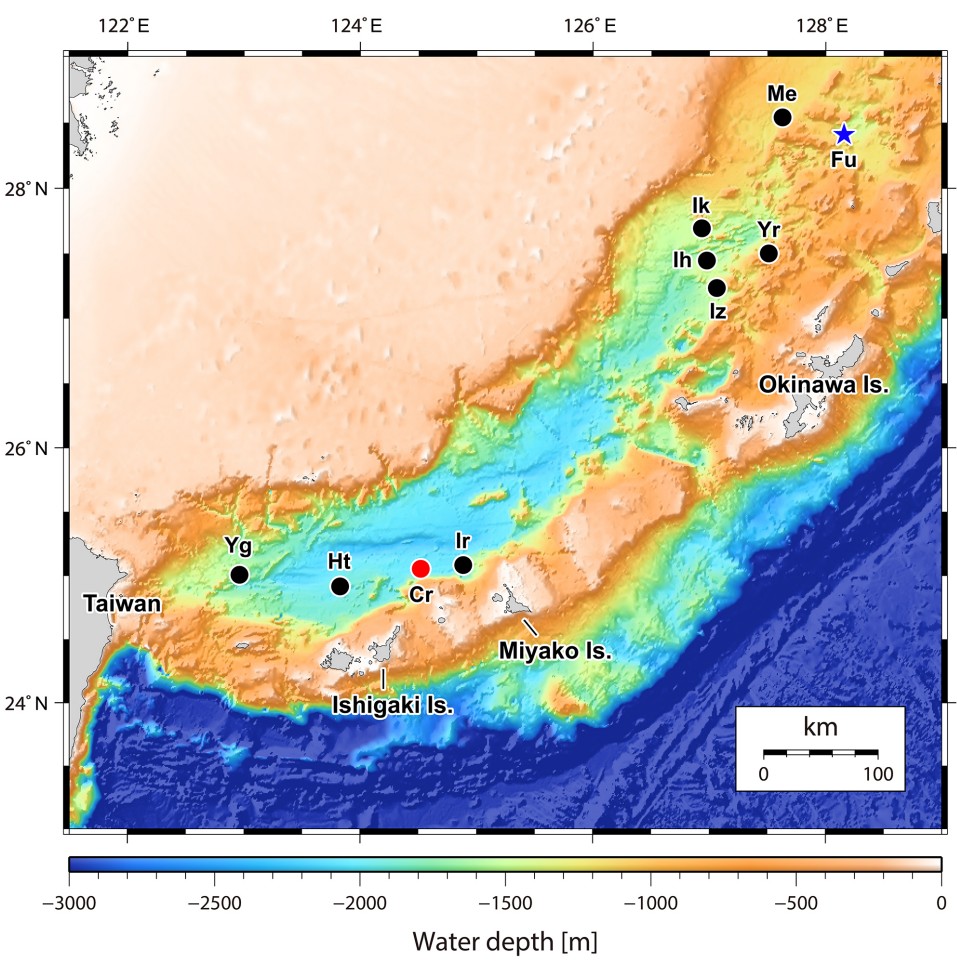

**Figure 1** **Map of the Okinawa Trough and surrounding areas showing locations of the two relevant hydrothermally influenced sites.** The blue star indicates Fukai site, Higashi-Ensei and the red dot indicates Crane site, Tarama Hill. Black dots indicate known vent sites in the Okinawa Trough (following *Watanabe & Kojima, 2015*). Site name abbreviations are as follows: Cr, Crane Site; Ht, Hatoma Knoll; Ih, Iheya Ridge; Ik, Iheya North Knoll; Ir, Irabu Knoll; Iz, Izena Hole; Me, Minami-Ensei Knoll; Yg, Daiyon-Yonaguni Knoll; Yr, Yoron Hole. Original 500 m mesh data originally obtained by Japan Coast Guard and publicly available from Japan Oceanographic Data Center (http://jdoss1.jodc.go.jp/vpage/depth500_file_j.html).

and was surveyed during ROV *KAIKO* dive #669 (for details see *Makabe et al., 2016*). A small weakly hydrothermally influenced site dominated by white branching Poecilosclerida sponges (*sensu Ise, 2012*), *Alaysia* sp. and *Lamellibrachia* sp. tubeworms, and *Bathymodiolus aduloides* Hashimoto & Okutani, 1994 mussels was found and named 'Crane' (Fig. 2C; *Makabe et al., 2017*).

### *Tristichotrochus ikukoae* (*Sakurai, 1994*)

One specimen of a calliostomatid gastropod was collected from Fukai site, Higashi-Ensei, Okinawa Trough (28°26.1064′N, 124°11.4889′E, 962 m deep, R/V *Kairei* cruise KR16-16, ROV *KAIKO* Dive #724, 2016/xii/04). Based on shell (diameter 22.6 mm, height 21.2 mm;

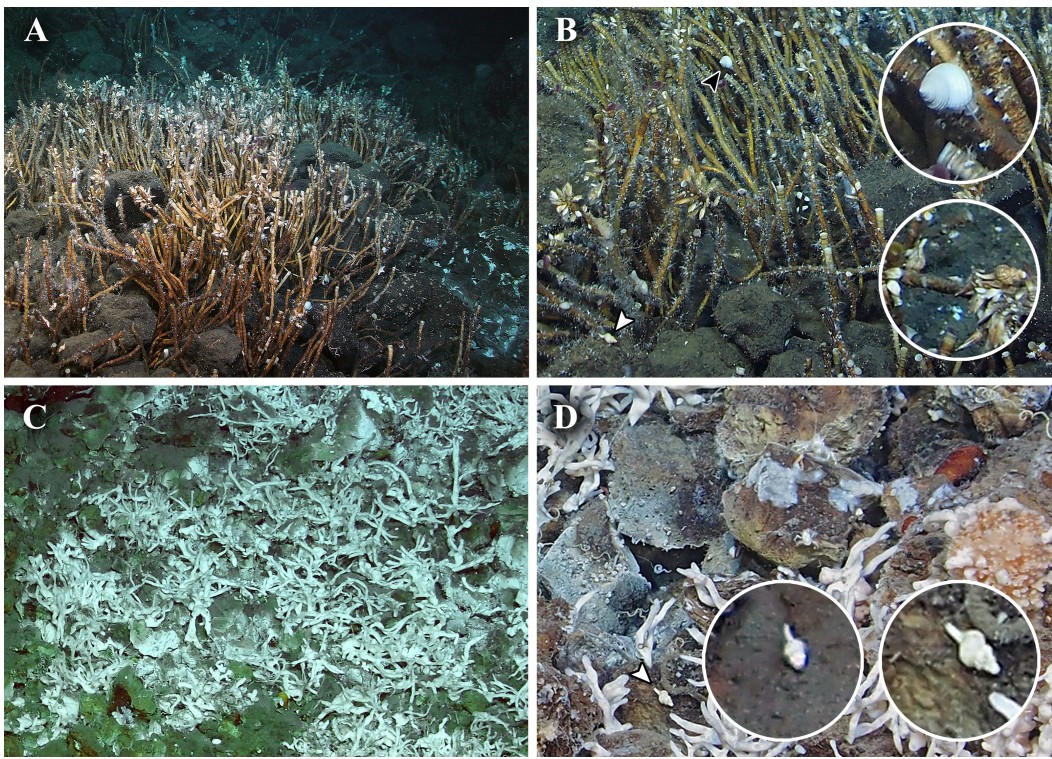

**Figure 2** *In situ* **photographs of the two hydrothermally influenced sites.** (A–B) Fukai site, Higashi-Ensei; (A) Overview, (B) Close-up; the black arrowhead indicates *Tristichotrochus ikukoae* and the white arrowhead indicates *Abyssotrophon* cf. *soyoae*. (C–D) Crane site, Tarama Hill; (C) Overview, (D) Close-up; the white arrowhead indicates *Abyssotrophon soyoae*. The cut-out photographs are magnified images of gastropods seen at each site.

Figs. 3A–3D) and radular (Fig. 4A) characteristics this specimen was clearly identifiable as *Tristichotrochus ikukoae* (*Sakurai, 1994*). To the knowledge of the authors, this is only the third time it has been formally reported and illustrated in the literature (*Higo, Callomon & Goto, 1999*; *Hasegawa, 2009*). A total of three individuals were sighted at the same site (Fig. 2B), living attached on *Lamellibrachia* sp. tubeworm tubes or hard surrounding substrata.

The taxonomic history of *Tristichotrochus ikukoae* is rather perplexing. It was initially described from Kochi Prefecture, Japan as *Otukaia ikukoae Sakurai, 1994*. Sakurai compared it to *Otukaia kiheiziebisu* (Otuka, 1939), commenting that *O. ikukoae* is 'smaller in size, and has obtuser ribs with larger granules' as well as 'has only 21 basal ribs in comparison' to *O. kiheiziebisu* (*Sakurai, 1994*). Later, *Okutani & Iwahori (1992)* reported *O. kiheiziebisu* from Tosa Bay, Kochi Prefecture, Japan which extended its distribution range from Sagami Bay. They reported that their specimens were variable in spiral rib development, with some specimens exhibiting five equally strong ribs. Shell and radulae of such 'unusual' specimens were illustrated (*Okutani & Iwahori, 1992*: Figs. 7, 10–12), and the shell was actually virtually identical to the holotype of *O. ikukoae*. Following this, *Hasegawa & Saito (1995)* considered *O. ikukoae* to be a junior synonym of *O. kiheiziebisu*, and Hasegawa (*Hasegawa, 2001*; *Hasegawa, 2009*) formally synonymised the two species. *Hasegawa*

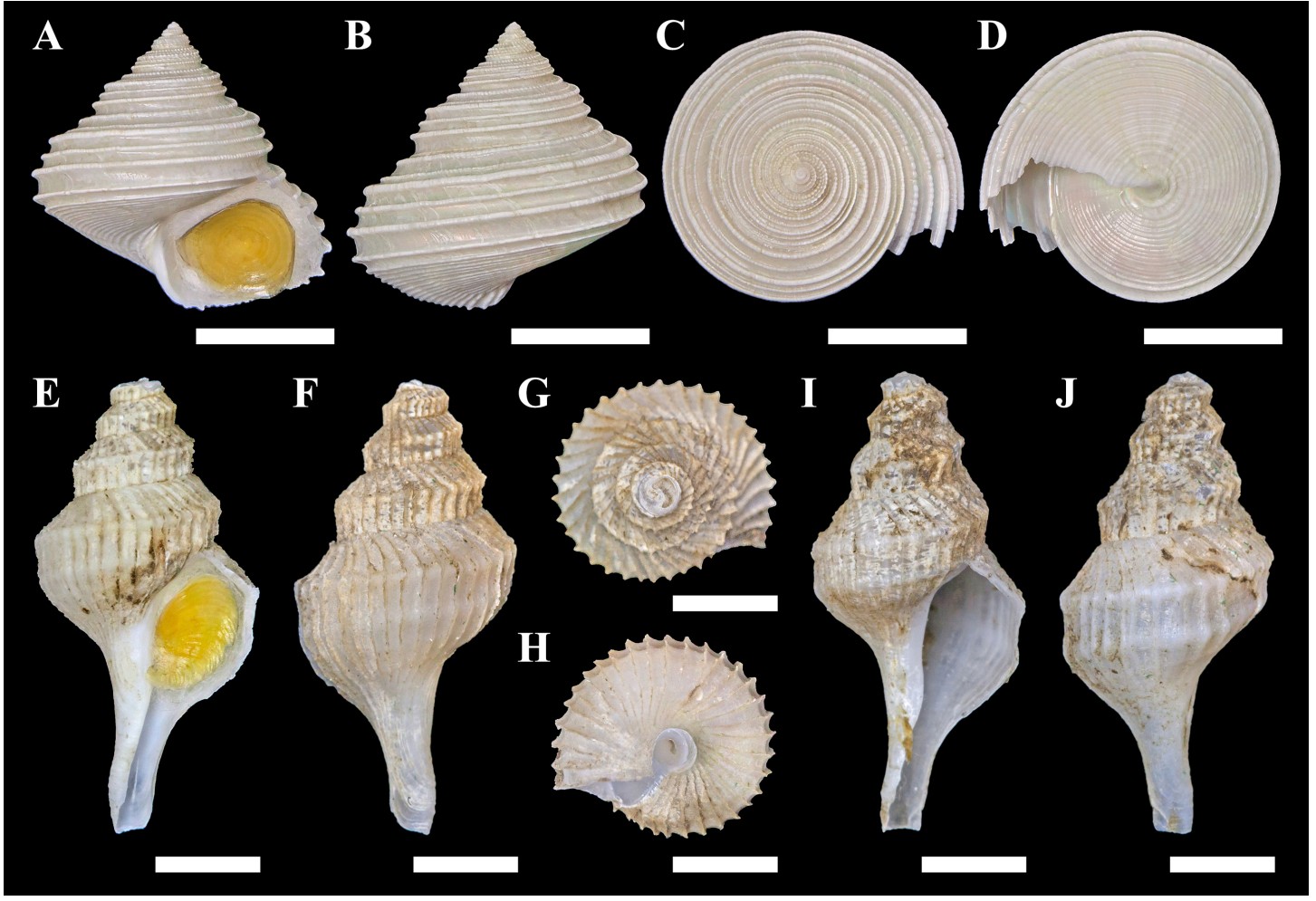

**Figure 3** **Specimens of the two gastropods collected from hydrothermally influenced areas.** (A–D) *Tristichotrochus ikukoae* from Fukai site, Higashi-Ensei. (E–J) *Abyssotrophon soyoae* from Crane site, Tarama Hill; (E–H) Specimen #1, (I, J) Specimen #2. Scale bars: (A–D) 1 cm, (E–J) 0.5 cm.

*(2001)* also reported *O. kiheiziebisu* from Tosa Bay, but due to the lack of illustration it is unclear whether specimens corresponding to the holotype of *O. ikukoae* were collected and examined. More recently, however, *Marshall (2016)* figured the radula of a typical *O. kiheiziebisu* and found it to be clearly distinct from that figured by *Okutani & Iwahori (1992)*. *Otukaia ikukoae* was reinstated as a valid species and transferred to *Tristichotrochus* based on its fewer lateral teeth compared to *Otukaia* (four to six compared to nine in *O. kiheiziebisu*).

Though the radula of *T. ikukoae* was reported to have only four pairs of lateral teeth (*Okutani & Iwahori, 1992*) compared to nine in *O. kiheiziebisu* (*Marshall, 2016*), this was questioned by *Marshall (2016)* who reasoned that there may actually be as many as five to six pairs. The specimen of *T. ikukoae* investigated in the present study has eight pairs of laterals (Fig. 4A), which is more numerous than *Marshall (2016)* predicted and makes the

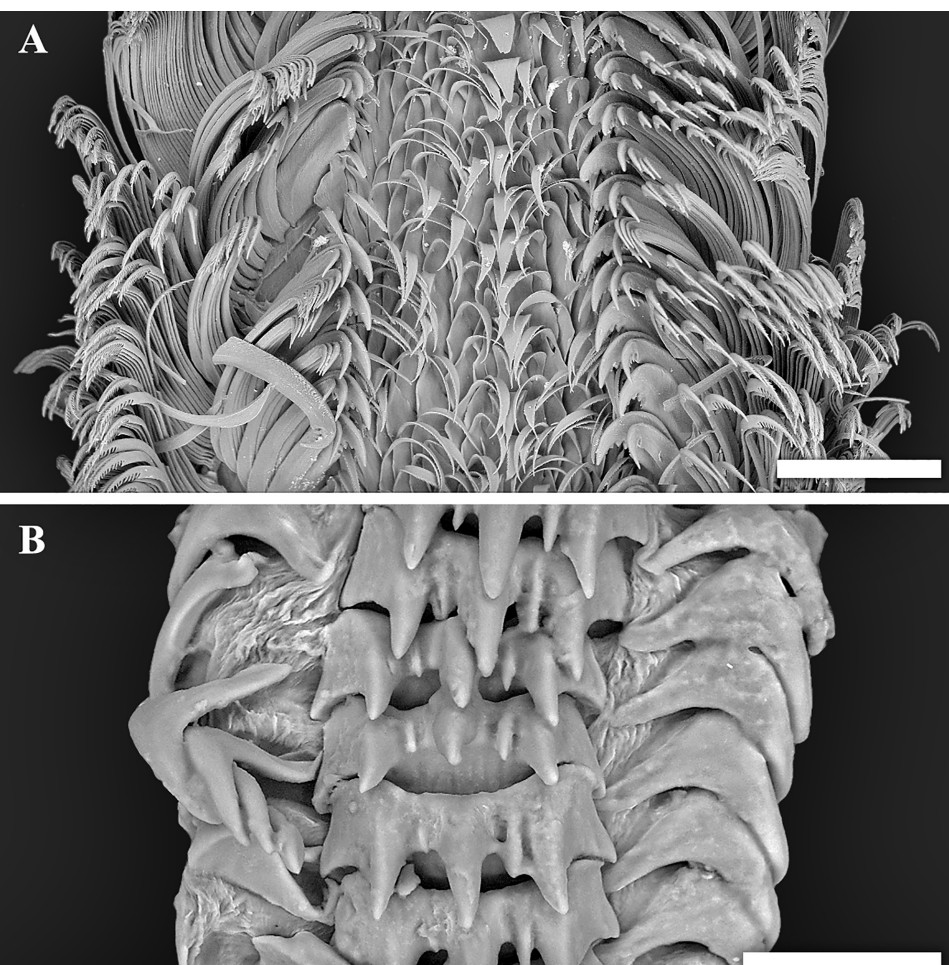

**Figure 4** **Scanning electron micrographs of radulae.** (A) *Tristichotrochus ikukoae* from Fukai site, Higashi-Ensei. (B) *Abyssotrophon soyoae* from Crane site, Tarama Hill. Scale bars: (A) 500 μm, (B) 50 μm.

radula formula ($n + 8 + 1 + 8 + n$). Due to only a single specimen being collected it is not possible to comment on the intraspecific variation in the radula of *T. ikukoae* at this point, but this is the most reliable account of the radula characteristics of this rare calliostomatid. Having as many as eight pairs of laterals means the radula of *T. ikukoae* actually fits well in genus *Otukaia*, and it is therefore returned to that genus herein, as *Otukaia ikukoae Sakurai, 1994*.

It should be noted that *O. ikukoae* and *O. kiheiziebisu* are in fact easily distinguished by shell characteristics as well, by the following two points: 1. *O. ikukoae* has five to six strong ribs on the body whorl whereas *O. kiheiziebisu* only has two to three (including the base keel); 2. The ribs of *O. ikukoae* are less sharply raised but more strongly beaded than those of *O. kiheiziebisu*. To assist future identification of these two species based on molecular barcodes, a COI barcode of the present specimen was sequenced and deposited in DNA Data Bank of Japan (DDBJ; accession number LC333137). A sequence of the same gene

is already available on GenBank for *O. kiheiziebisu* (from *Williams et al., 2010*) under the accession number AB505274, the identity of the individual sequenced has been ascertained and confirmed as *O. kiheiziebisu* (T. Nakano pers. comm., in (*Marshall, 2016*)). Based on comparison between these two sequences, the pairwise difference between the two species is 5.8%, which is higher than the average of 3–4% between closely related species in marine gastropods (*Meyer & Paulay, 2005*).

The size of the present specimen (shell height 21.2 mm) is just slightly smaller than previous reports as shell height of the holotype is 23.9 mm (*Sakurai, 1994*; but noted as 23.6 mm in *Higo, Callomon & Goto, 2001*) and height of specimens definitively referable to *O. ikukoae* reported by *Okutani & Iwahori (1992)* range between 23.5 mm–26.7 mm.

Since *O. ikukoae* has not been recorded from outside Kochi Prefecture (*Higo, Callomon & Goto, 1999*; *Hasegawa, 2009*), the present new record extends its distribution range to the southwest by about 700 km. The collection depth of 962 m is deeper than the Tosa Bay record (700 m; *Okutani & Iwahori, 1992*), and since the depth that the type specimen was collected from is not known (*Sakurai, 1994*), this represents the deepest record for the species.

### *Abyssotrophon soyoae* (*Okutani, 1959*)

Two specimens of trophonine muricids were collected from the Crane site, Tarama Hill, Okinawa Trough (25°4.5388′N, 124°31.0103′E, 1973 m deep, R/V *Kairei* cruise KR15-16, ROV *KAIKO* Dive #669, 2015/x/29). A total of five individuals was sighted at this site (Fig. 2D). The morphology of the two specimens matched best, in terms of shell (specimen #1, height 22.2 mm, width 11.0 mm, Figs. 3E–3H; specimen #2, height 22.5 mm, width 10.5 mm, Figs. 3I and 3J) and radular (specimen #1, Fig. 4B) characters, with *Abyssotrophon soyoae* (*Okutani, 1959*). Originally described from off Miki-Saki, Kumano-nada, Kii Peninsula, Japan (33°48.8′N, 136°34.8′E) as *Trophonopsis soyoae* Okutani, 1959, this is only the fourth time it has been recorded formally (*Okutani & Iwahori, 1992*; *Higo, Callomon & Goto, 1999*; *Hasegawa, 2001*; *Tsuchiya, 2017*). As *Hasegawa (2001)* indicated, the record from Russia (*Egorov, 1993*) is unlikely to be an authentic record of this species due to significant differences in morphology. Furthermore, three individuals that appeared to belong to the same species were seen in Fukai site (Fig. 2B, the same location as *O. ikukoae* collecting data above), but unfortunately none were collected.

Conchologically *A. soyoae* is easily recognised by two strong keels combined with very crowded axial ribs (*Tsuchiya, 2017*). One significant difference between the present material and previously reported specimens is that the axial ribs are less crowded (about 35 vs about 70; *Okutani & Iwahori, 1992*). However, there appears to be some variability in rib frequency across life stages (see Fig. 3J; *Okutani & Iwahori, 1992*: Fig. 28) and the authors consider this difference to be intraspecific, rather than interspecific, variability until further evidence becomes available proving otherwise. The specimens examined herein also have higher spires than the holotype (*Okutani, 1959*), but they match well with specimens previously reported from Tosa Bay, Kochi Prefecture, Japan (*Okutani & Iwahori, 1992*). The shell height to width ratios were 2.02 and 2.14, which gives an impression that they are more stout than the previous specimens which range between 2.11–2.69 (2.11 in the

holotype; *Okutani & Iwahori, 1992*) but only because the earlier whorls are much more corroded in the present specimens, likely due to the acidic hydrothermal environment. The radula is very much the same as imaged by *Okutani & Iwahori (1992)*, including the formula $(1+1+1)$ and in having a single middle cusp on the central tooth. Although the soft parts of the specimens investigated were not of good quality, eyes were present in both individuals and the smaller individual (specimen #1, male) possessed a very large penis as previously reported (*Okutani & Iwahori, 1992*). To facilitate future identification and genetic comparison, the COI barcode sequence was obtained from specimen #1 and deposited in DDBJ under accession number LC333138. The closest match available on GenBank is a sequence of *Boreotrophon truncatus* (Strøm, 1768), accession number HQ919183 (from *Layton, Martel & Hebert, 2014*) with 9% pairwise difference.

Since this species has only been reported from the type locality and Tosa Bay (*Higo, Callomon & Goto, 1999*; *Hasegawa, 2001*; *Tsuchiya, 2017*), the present record extends its distribution range over 1,250 km to the southwest. The collection depth of the present record (1,973 m) is within the bathymetric range between the type locality (2,050 m; *Okutani, 1959*) and the Tosa Bay records (700–835 m; *Okutani & Iwahori, 1992*; *Hasegawa, 2001*).

## Significance of the new records from vent ecosystems

Trochoid gastropods are commonly found in many hydrothermal vent sites around the globe (e.g., *Iheyaspira*, *Cantrainea*) but calliostomatids have only been recorded twice (*Sasaki et al., 2010*). These include *Falsimargarita nauduri* (*Warén & Bouchet, 2001*) from the 17° S hydrothermal vent site, East Pacific Rise (*Warén & Bouchet, 2001*) and *Maurea chilena* (Rehder, 1971) from Concepción methane seep off Central Chile (*Sellanes, Quiroga & Neira, 2008*). Although *Sellanes, Quiroga & Neira (2008)* also reported *Otukaia crustulum* Vilvens & Sellanes, 2006 from the Concepción methane seep, this species has been moved to Calliotropidae, as *Calliotropis crustulum* (*Marshall, 2016*). The present report of *Otukaia ikukoae* is therefore the second record of calliostomatids from vent ecosystems and the third record from chemosynthetic ecosystems.

Muricids are found from intertidal waters down to over 7,000 m deep, and *Abyssotrophon* is one of the deepest-living genera occurring even over 7,200 m deep in the case of *Abyssotrophon hadalis* (Sysoev, 1992) (*Egorov, 1993*). Five species of Muricidae are known from Concepción methane seep off Central Chile (*Sellanes, Quiroga & Neira, 2008*) and two are known from methane seeps in Barbados Prism (*Warén & Bouchet, 2001*), making a total of seven records from hydrocarbon seeps. Except *Enixotrophon concepcionensis* (*Houart & Sellanes, 2006*) which is in the subfamily Pagodulinae (*Houart & Sellanes, 2006*), all other six species recorded are in the subfamily Trophoninae (genera *Trophon* and *Coronium*; *Sasaki et al., 2010*). The present record of *Abyssotrophon soyoae* is the third record of this family from vent ecosystems and the tenth from chemosynthetic ecosystems. Both families were previously unrecorded from chemosynthetic ecosystems of not only the Okinawa Trough but the western Pacific as a whole (*Sasaki et al., 2010*; *Watanabe & Kojima, 2015*).

Although the two species reported herein were found in hydrothermally influenced settings, they are also known from regular deep-sea bottom (*Okutani & Iwahori, 1992*) and

thus not endemic to chemosynthetic ecosystems. Neither new hydrothermally influenced site reported herein is vigorously active, and is instead more like 'hydrothermal seeps' (*Levin et al., 2012*) than conventional high-temperature fluid venting systems. The presence *O. ikukoae* and *A. soyoae* in weakly diffuse flow areas indicate that they are able to tolerate some levels of environmental stresses associated with hydrothermal ecosystems (e.g., increased acidity, concentration of toxic substances and heavy metals), allowing them to invade and benefit from the high productivity of hydrothermal vent ecosystems.

While feeding behaviour was not observed in the field, the ecological roles the two species play may be inferred as follows. Deep-water calliostomatids are mostly carnivorous grazers feeding on cnidarians or sponges (*Marshall, 1995*), and from the fact that they were observed living on tubeworm tubes *O. ikukoae* in Fukai site probably graze on sponges and/or biofilm growing on the tubes. Muricids are carnivores feeding on other marine invertebrates (*Merle, Garrigues & Pointier, 2011*), and *A. soyoae* is unlikely to be an exception. In both localities the *Abyssotrophon* individuals were seen in close proximity to other molluscs (e.g., *Bathymodiolus aduloides* at Crane site) and barnacles (e.g., *Ashinkailepas seepiophila* at Fukai site) which the *Abyssotrophon* likely prey upon. In interacting with vent endemic species in such ways, *O. ikukoae* and *A. soyoae* may contribute to trophic transfer of chemosynthetically derived production from vents to surrounding ambient environment, which some other predators such as seastar, fish, and octopus have been suggested to play a role in (*Levin et al., 2016* and references therein). The apparent absence of these two species from other Okinawa Trough vents is possibly an artefact caused by the relatively little effort put into exploring the peripheral areas.

In fact, non-vent endemic deep-water predatory gastropods have been recorded from vent periphery in several occasions, for example *Buccinum viridum* Dall, 1889 on the Juan de Fuca Ridge (*Tunnicliffe & Fontaine, 1987*; *Sarrazin et al., 1997*), *Enixotrophon carduelis* and *E. obtusus* on the Kermadec Ridge (*Marshall & Houart, 2011*), and *Neptunea robusta* Okutani, 1964 in the Okinawa Trough (*Watanabe & Kojima, 2015*). Their success may be a combined effect of the ability to tolerate moderate environmental stress from vent periphery and the fact that few vent endemic gastropod species are predators, meaning there is an available niche for these predatory species given the high productivity. In the Okinawa Trough vents, for instance, only one vent endemic predatory gastropod, *Thermosipho desbruyeresi* (Okutani & Ohta, 1993), is known (*Watanabe & Kojima, 2015*).

The present records signify that outskirts of hydrothermally influenced areas likely provide important habitats and energy resources for some non-endemic species (*Levin et al., 2016*). These peripheral areas may be important for the evolution of biological adaptations to chemosynthetic ecosystems by providing the intermediate environment between 'extreme' vents and the regular surrounding seafloor.

## CONCLUSIONS

Exploration of two new hydrothermally influenced ecosystems in the Okinawa Trough led to the collection of two gastropod snails previously unknown from chemosynthetic ecosystems. One was *Otukaia ikukoae*, the second calliostomatid ever recorded from vents

and the third record of the family for chemosynthetic ecosystems overall. The other was the muricid *Abyssotrophon soyoae*, marking the third time Muricidae has been found at hydrothermal vents and the tenth record at chemosynthetic ecosystems. Both species are extremely rare and this report represents the third record for *O. ikukoae* and the fourth for *A. soyoae* in the literature, as well as the first time they are seen in their natural habitat. Neither family has been recorded at chemosynthetic ecosystems in the western Pacific. Since both species are also known from regular non-chemosynthetic deep-sea and they were recovered from areas with only weak diffuse flow venting, they have most likely invaded vent environment from regular sea bottom and play a role in trophic transfer of chemosynthetic production to the surrounding seafloor. The fact that such species are able to tolerate (some) influence from hydrothermal vents signify that such 'outskirt' areas may provide key habitats for them, and that these areas may play a role in the evolution of biological adaptations to 'extreme' chemosynthetic ecosystems.

## ACKNOWLEDGEMENTS

The authors would like to express their sincere gratitude to the Captain and crews of R/Vs *Kairei* and *Yokosuka*, as well as the operation team of the ROV *KAIKO*, for their untiring support during the relevant cruises (YK14-16, KR15-16, YK16-07, KR16-16). Kyoko Okino (AORI, the University of Tokyo) is gratefully acknowledged for leading the YK14-16 cruise, and Kentaro Nakamura (the University of Tokyo) for analysing the MBES data which led to the discovery of the new hydrothermal sites. Roland Houart and Christopher Moe are thanked for their valuable aid in identifying the *Abyssotrophon*. Kazuya Kitada (JAMSTEC) kindly provided the base map for Fig. 1 used herein. Bruce Marshall (Museum of New Zealand Te Papa Tongarewa) and Verena Tunnicliffe (University of Victoria) are thanked for providing comments on an earlier version which significantly improved the manuscript. The Fukai site was named in reference to a poisonous jungle of the same name in the film "Nausicaä of the Valley of the Wind" (Studio Ghibli, 1984); the director Hayao Miyazaki and Studio Ghibli are gratefully acknowledged for being a great source of inspiration.

### Funding

This study was supported by Council for Science, Technology, and Innovation (CSTI) as the Cross Ministerial Strategic Innovation Promotion Program (SIP), Next-generation Technology for Ocean Resource Exploration. The genetic sequencing was funded by a Japan Society for the Promotion of Science KAKENHI (grant no. 15K18602) awarded to Hiromi Kayama Watanabe. The funders had no role in study design, data collection and analysis, decision to publish, or preparation of the manuscript.

### Grant Disclosures

The following grant information was disclosed by the authors:
Council for Science, Technology, and Innovation (CSTI).
Japan Society for the Promotion of Science KAKENHI: 15K18602.

## Competing Interests

The authors declare there are no competing interests.

## Author Contributions

- Chong Chen conceived and designed the experiments, performed the experiments, analyzed the data, contributed reagents/materials/analysis tools, wrote the paper, prepared figures and/or tables, examined and identified the specimens morphologically, participated in the relevant research cruises.
- Hiromi Kayama Watanabe performed the experiments, analyzed the data, contributed reagents/materials/analysis tools, reviewed drafts of the paper, aided the identification of vent fauna.
- Junichi Miyazaki and Shinsuke Kawagucci contributed reagents/materials/analysis tools, reviewed drafts of the paper, participated in the relevant research cruises.

## DNA Deposition

The following information was supplied regarding the deposition of DNA sequences:

The newly generated COI barcode sequences are deposited in DNA Data Bank of Japan with accession numbers LC333137 (*O. ikukoae*) and LC333138 (*A. soyoae*).

## Data Availability

Specimens studies in this study are deposited and accessioned at JAMSTEC marine biological sample collection; Otukaia ikukoae: JAMSTEC No. 1160052681; Abyssotrophon soyoae: JAMSTEC No. 1150046767. The newly generated COI barcode sequences are deposited in DNA Data Bank of Japan with accession numbers LC333137 (*O. ikukoae*) and LC333138 (*A. soyoae*).

## Supplemental Information

Supplemental information for this article can be found online at http://dx.doi.org/10.7717/peerj.4121#supplemental-information.

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
