# Peer review of "Unanticipated discovery of two rare gastropod molluscs from recently located hydrothermally influenced areas in the Okinawa Trough"

_PeerJ, doi:10.7717/peerj.4121_

## Round 0.1 · original submission · Minor Revisions

I agree with the reviewers' comments and (as also expressed by Reviewer 2) would like to see some additional discussion concerning the intersection of vent and non-vent species. I look forward to receiving the revised version!

·

Basic reporting

See attached report.

Experimental design

See attached report.

Validity of the findings

See attached report.

Additional comments

See attached report.

·

Basic reporting

The manuscript language is good both in overall structure and specific use. Literature citation is mostly adequate although limited (suggestions below). Figures are fine.
Fasta sequences were provided but I don’t see them in DDBJ.

Experimental design

It’s an observational study. Methods are adequate.

Validity of the findings

It is fine to draw attention to these records at chemosynthetic systems. However, as neither two species themselves nor the presence of the family at vents is new, I would suggest the authors make a larger contribution by:
i) Drawing a better picture of the likely role these animals play in this ecosystem (grazer? predator?)
ii) Comment on the intersection of vent and non-vent species. There is a fairly robust literature that details the peripheral vent environment – this record is an important contribution so set it in that context. The most recent review, I believe, is Levin et al. 2016. 2016. Hydrothermal vents and methane seeps: Rethinking the sphere of influence. Frontiers in Marine Science, 3, dx.doi.org/10.3389/fmars.2016.00072 There are quite a few instances of deep-water gastropods on the periphery of vents and seeps – why are they successful in this niche? (I see Buccinum viridum frequently among tubeworms in the northern Juan de Fuca Ridge).

Additional comments

Do you think they are competing with species that use the resources in a similar fashion? Or are these available niches? Why are they not found at the other vents in the region?
Suggest that the last sentence of the Introduction is part of the habitat characterization and should move to the Discussion to explain the presence of these taxa. …. and when I get to end of Discussion, I do not understand this “maybe a seep” comment. The best indication of whether it is a cross-over habitat is your list of species – how do they compare with Watanabe et al’s list of vent fauna in the region? Are there seep fauna here, too? Are there any supporting temperature data to decide if there is a thermal anomaly here?

Other points
L 62: “drastically”? Are you talking about end-member fluids? pH in waters around animals is often no different from seawater. But you can justify if the Okinawa region is notably different in diffuse fluids.
l 69: Lutz and Kennish do not really test this idea – but this paper does: Van Dover CL, Trask JL. 2000. Diversity at deep-sea hydrothermal vent and intertidal mussel beds. Marine Ecology Progress Series 195:169-178.
l 90: in English, “Neither” takes singular subject and verb.
l. 104: remove ‘by the junior author.’ Are there any temperature or chemical measurements made?
Good imagery of T. ikukoae. Is the size within the range originally reported for the species?
Figure 1: Add N and E to lat/long; attach two or three island names to orient the reader. Also, I think it useful to indicate the other vent sites from Watanabe and Kojima 2015.

---

## Round 0.2 · accepted · Accept

Thank you for your revised manuscript. I fully appreciate your response to the referee comments and am now happy to move this forward into production.